# The many faces of compensation: The similarities and differences between social and facial models of perception

Mathias Schmitz *, Antoine Vanbeneden, Vincent Yzerbyt

Université Catholique de Louvain, Institute for Research in the Psychological Sciences, Louvain-la-Neuve, Belgium

* mathias.schmitz@pucp.pe

## Abstract

Previous research shows that stereotypes can distort the visual representation of groups in a top-down fashion. In the present endeavor, we tested if the compensation effect—the negative relationship that emerges between the social dimensions of warmth and competence when judging two social targets—would bias the visual representations of these targets in a compensatory way. We captured participants' near spontaneous facial prototypes of social targets by means of an unconstrained technique, namely the reverse correlation. We relied on a large multi-phase study ($N = 869$) and found that the expectations of the facial content of two novel groups that differed on one of the two social dimensions are biased in a compensatory manner on the facial dimensions of trustworthiness, warmth, and dominance but not competence. The present research opens new avenues by showing that compensation not only manifests itself on abstract ratings but that it also orients the visual representations of social targets.

**Data Availability Statement:** The data and R scripts to carry out the analyses are publicly available on Open Science Framework: https://osf.io/hk8av/.

## Introduction

Dimensional compensation takes place when a target comes across as superior to another on one of the two fundamental dimensions of social perception, i.e., warmth or competence, but as inferior on the other dimension ([1]; for a review, see [2]). Notwithstanding the fact that this effect has proven robust at an indirect, explicit, and verbal level, no study has tested if this phenomenon could also emerge in people's representation of faces. This is all the more important given the key role of facial cues in the context of social interactions and the top-down impact of stereotypes on facial representations [3]. On this basis, we tested if compensation could bias the visual rendering of social targets in a compensatory manner.

### The dimensional compensation effect

Almost a century ago, Lippman [4] referred to stereotypes as "pictures in our heads." This conceptualization emphasizes the visual component of stereotypes while remaining closely related to the contemporary definition of stereotypes as beliefs or opinions about the

**Funding:** The authors disclosed receipt of the following financial support for the research, authorship, and/or publication of this article: This work was supported by the Fonds National de la Recherche Scientifique (FNRS/FSR) (grant number 1.A393.17) awarded to Mathias Schmitz. The funders had no role in study design, data collection and analysis, decision to publish, or preparation of the manuscript.

**Competing interests:** The authors have declared that no competing interests exist.

characteristics, attributes or behaviors shared by individuals from the same group [5]. A long tradition of research has shown that, although the collection of traits used to describe groups could in principle be vast and diverse, the content of stereotypes revolves around two fundamental dimensions ([6–9]; for a review, see [10]). Different labels have been proposed for these two dimensions depending on the theoretical frameworks, and sub-dimensions or facets have also been identified ([11–13]; for a review, see [14]). On the one hand, the warmth/communion dimension conveys information about the target's intention (i.e., "is it a friend or a foe?") and comprises the facets of friendliness (e.g., nice, friendly, and caring) and morality (e.g., honest, moral, and trustworthy). On the other, the competence/agency dimension is inferred from the target's capacity to concretize his or her intentions and encompasses the facets of ability (e.g., skilled, efficient, and organized) and assertiveness (e.g., determined, ambitious, and self-confident).

Interestingly, most stereotypes come across as positively valenced on one dimension but negatively on the other—the so-called mixed or ambivalent stereotypes (e.g., gender stereotypes [15]). Yzerbyt et al. [1] have suggested that such an arrangement may stem from the compensation effect that materializes in the negative relationship between the two dimensions when judging two targets. For instance, in the context of national groups, Italians were perceived as warmer but less competent than Belgians ([16]; see also [17]). In a more controlled setting, when asked to judge two novel groups with one group initially presented as more competent than the other, participants compensated for this difference in the opposite direction on the warmth dimension [18].

A large body of research has shown not only the robustness of the compensation effect but also identified a series of boundary conditions for its emergence (e.g., [18–21]; for a review, see [2]). Still, many tested the compensatory relation between warmth and competence under constrained conditions, that is, participants only had a limited set of possible responses options that the researchers had selected beforehand (e.g., list of traits, labels, and possible answers to questions). Moreover, compensation was only tested at a verbal level. For instance, by the means of traits' ratings (e.g., [22]) or by having people choose specific content- and valence-oriented questions [23], select behavioral descriptions [24], or react to associations between group labels and traits (i.e., Brief-IAT [25]). Hence, the question remains if non-verbal stimuli, such as visual information conveyed by faces, may also be biased in a compensatory way and under unconstrained conditions. This issue becomes even more relevant when considering the role and importance of facial information in everyday life social interactions [3, 26].

## Impressions from faces

Faces are one of the most common visual and rich components of our social environment and are imbued with essential information for navigating everyday life social interactions. Not surprisingly, people are strongly drawn to make a myriad of social inferences from facial features [27, 28] despite little evidence for their accuracy (for a review, see [3]). These facial impressions can have consequential outcomes such as political preferences and voting intentions [29–31] or sentencing decisions [32].

Earlier research on the structure underlying social judgments from faces revealed that two core dimensions could be identified [33, 34]. The first axis was best associated with valence/trustworthiness judgments pointing to the target's intention (i.e., harmless vs. harmful), while the second axis was approximated by dominance judgments indicating the target's capability to carry out these intentions. According to the overgeneralization hypothesis [35–38], impressions from neutral faces are overgeneralization of cues that provide adaptive information. Trustworthiness would be an overgeneralization of emotional cues (e.g., anger or happiness)

signaling approach or avoidance tendencies [33, 34, 39, 40], whereas dominance judgments would stem from overgeneralizations of facial indicators of physical strength and facial maturity [33, 41]. This two-dimensional model was replicated more recently with the addition of a third dimension related to youthful-attractiveness judgments that could serve sexual selection functions [42–45]. More recent efforts have begun to investigate the universality of this model (see [43, 46–49]).

The eye-catching resemblance between the competence-by-warmth social model and the dominance-by-trustworthiness facial model was investigated by Sutherland et al. [50]. Their results pointed to substantial overlap between judgments on the two horizontal dimensions of trustworthiness and warmth while the relation between the vertical dimensions of dominance and competence was weaker. These authors suggested different explanations for the intriguing discrepancy regarding the second axis. For instance, cues signaling competence may vary between the conceptual and facial level, that is, some components that are diagnostic of competence may be more readily encoded in faces (e.g., physical strength and dominance) than others (e.g., intelligence and skill). The latter distinction resonates with recent work on the facets of assertiveness and ability, respectively Another possibility, based on an evolutionary perspective, is that competence/prestige (i.e., sharing expertise to gain respect) and dominance (i.e., the use physical strength to induce fear) represent two different strategies—the former being beneficial and the later harmful for others—to climb the social ladder [51, 52].

## Visual representation of faces

The above review reveals that people are inclined to make social inferences from faces. An interesting question is whether the reverse process also takes place. Specifically, can our previous knowledge (e.g., stereotypes), beliefs (e.g., political ideologies), or sympathy (e.g., prejudice) for a given group bias the way we imagine their faces? A first response comes from Eberhardt et al.' [53] study in which participants were presented with a picture of a racially ambiguous target before drawing the target's face. The drawings were clearly biased by the racial label attached to the target. Building on these initial findings, several studies showed that visual representation of social targets can be biased in a top-down fashion by *a priori* beliefs, such as racial stereotypes (e.g., [54–56]), minorities stereotypes [57, 58], national stereotypes (e.g., [59, 60]), gender stereotypes [61], or occupational stereotypes [26, 57, 62, 63]. Moreover, behavioral information [64] or minimal group membership [65] proved sufficient to bias the representation of facial trait-diagnostic-features of novel groups.

Most of the above efforts relied on a variant of the reverse correlation technique (see [54, 66, 67]) to capture the visual representation of the target and identify the specific visual diagnostic features that drive social inferences. In a typical reverse correlation task (RC), participants go through several trials for which they must choose between a pair of noisy faces the face that looks the most like the specified target (e.g., "the most Moroccan-looking face"). These noisy faces are random variations of a single face (i.e., the base-face). The average of all selected noise patterns superimposed to the base-face is then used to produce the classification image (CI)—a visual read-out of the target mental representation [68].

This procedure offers several advantages (for a review, see [68]; but see also [69]). First and foremost, it is unconstrained and data-driven. Indeed, the variation in the stimulus set—the noisy faces generated from a random noise pattern superimposed to the base-face—is immensely large and can thus span an entire space of hypotheses rather than a single one. Secondly, it allows visualizing "near *spontaneous use of information* because participants are free to adopt whatever criteria they want for their judgments (in fact, participants might not even be aware of the criteria they adopt)" ([68], p. 336). In sum, this method clearly contrasts with

more traditional social judgment paradigms or indirect measures that force participants to make judgments or categorizations along pre-established dimensions or categories rooted in researchers' *a priori* hypotheses that may not be germane to real-life scenarios [26, 68, 69].

Using this reverse correlation approach, previous studies successfully established the encoding and decoding of the trustworthiness-by-dominance and the warmth-by-competence dimensions' content onto physical facial cues. Specifically, the trustworthiness and dominance classification images (CIs) were negatively correlated, whereas the warmth and competence CIs were positively correlated (e.g., [26, 66, 70, 71]). In addition, trustworthiness and warmth CIs were highly similar, and both overlapped with judgments of valence ([71]; see also [33]). Interestingly, there was a dissimilarity between the dominance and competence CIs driven by the negative evaluation of the former and positive evaluation of the latter. In particular, Oliveira et al. [72] found that valence (which overlaps with warmth/sociability) had a negative linear relationship with dominance in the domain of face representation but had a U-shaped relation when judging trait concepts (see also [73]; for a more general argument, see [74]).

### The present research

Research on the compensation effect has mainly taken place at a verbal level (e.g., traits' ratings) and under constrained conditions (e.g., a limited number of items or stimuli). In light of these limitations, we aimed to examine whether compensation would also emerge at a visual level and in a more spontaneous way. Previous work in the domain of face perception suggests that competence and warmth are positively related (i.e., a halo effect) at both the judgmental and representational level [26, 50, 71]. However, these studies did not aim to test compensation *per se*, nor did they create an appropriate setting (e.g., comparison context; see [2]). The emergence of a visual correlate of the compensation effect would therefore constitute convincing evidence, especially in the light of the likely emergence of a halo effect already documented.

The research was conducted in four phases. In phase 1 and 2, we tested if the compensation effect would be present in the facial representation of two social targets. To do so, we presented participants with two fictitious groups that differed on one of the two fundamental dimensions (competence in phase 1; warmth in phase 2) but were equal on the other (warmth in phase 1; competence in phase 2). The use of new groups as targets provides a more controlled setting and precludes the possibility of a perceptual bias caused by previous knowledge. We then relied on the reverse correlation procedure to capture participants' visual representation of the two targets. Independent judges then rated these visual renderings on the two fundamental dimensions of social perception. In both phases, we predicted compensatory ratings of the prototypical faces. Because past findings have pointed out some divergence between the two models of social and face perception, in phase 3, we asked a new sample of judges to rate the previously produced faces on trustworthiness and dominance. We also predicted a negative relationship between these ratings. In phase 4, we sought to replicate and further validate our findings by manipulating the design used to collect the data (i.e., within- vs. between-participants).

In this research, we relied on the standard implementation of the reverse correlation paradigm [54]. Given that there exists no reliable method (see [68]) to estimate the number of participants (hereafter referred to as "producers") required for a reverse correlation task, we followed previous studies recommendation ([66]; see also [71]) of at least twenty producers per condition. Likewise, previous studies have relied on a wide variety of samples sizes ranging between 31 and 400 participants (hereafter referred as "judges") to rate the visual outcomes generated from the RC task (e.g., [71, 75, 76]). The final sample sizes in the various phases of

our study fulfilled the minimal number of producers and judges widely recommended and we increased these numbers in light of available financial resources and participants' pool availability.

## Materials and methods

### Participants

The current study involved multiple independent samples of participants. Next to "producers" who performed the reverse correlation (RC) task and produced classification images (CIs), "judges" rated these images. Phase 1 involved 136 producers from the United States ($M_{age}$ = 37.69, $SD_{age}$ = 14.00, 79 women, 57 men) recruited online via a crowdsourcing platform, Prolific Academic, and 59 judges from a Belgian French-speaking university (demographic data from this sample was lost due to a programming error). Phase 2 comprised 66 producers from a Belgian French-speaking university ($M_{age}$ = 21.97, $SD_{age}$ = 2.47; 51 women, 15 men) and 105 judges from the United Kingdom sampled through Prolific Academic ($M_{age}$ = 35.86, $SD_{age}$ = 11.84; 68 women, 36 men, one other). In phase 3, we recruited 201 judges from the UK through Prolific Academic ($M_{age}$ = 36.18, $SD_{age}$ = 12.94; 138 women, 63 men), whereas 302 judges ($M_{age}$ = 35.42, $SD_{age}$ = 11.46; 233 women, 69 men) from the same population took part in phase 4. All participants agreed to an informed consent form (online) prior to their participation.

### Materials

We selected six male faces with similar shapes from the Radboud Face Database [77] and split these into two sets. We generated four noisy faces from each face by superimposing a random noise pattern with the default settings from the *rcirc* package [78]. We used the resulting noisy faces in the impression formation phase as faces exemplars of group members (similar to Dotsch et al. [64]). Next, we morphed the six male faces previously selected into a single base image using Psychomorph [79]. We then generated three hundred and fifty pairs of noisy faces with the *rcirc* R package (default settings) by adding or subtracting a noise pattern from the base image (see [66, 67]) as illustrated in Fig 1. The stimuli materials had a 512 × 512 pixels resolution.

### Measures

In phases 1 and 2, we captured producers' face representation of target groups, also known as classification images (CIs), with a reverse correlation (RC) variant, namely the two images forced classification task [54]. The literature shows that previous RC studies relied on 300–500 trials, which is large enough to generate reliable group-wise but not individual CIs [75]. Since our focus was on the group-wise CIs, we opted for 350 trials. In each trial, participants had to "choose the face that looked most like a green [blue] group member" between two adjacent noisy faces. The resulting CI of a given target group was obtained by averaging all the noise patterns selected by all the participants and then superimposing this noise pattern to the base image. We evaluated phase 1 producers' explicit impressions about the two groups through traits rated on a 7-point scale (0 = strongly disagree; 6 = strongly agree). We used competent, efficient, capable, and intelligent to assess the competence dimension, and sociable, nice, warm, and caring for the warmth dimension (all $\alpha$'s ≥ .90). In phase 2 we increased the spectrum of traits used to rate the visual presentations to the four facets of the two fundamental dimensions. We used traits related to ability (competent, efficient, and hardworking) and assertiveness (ambitious, determined, and self-confident) for the competence dimension, and

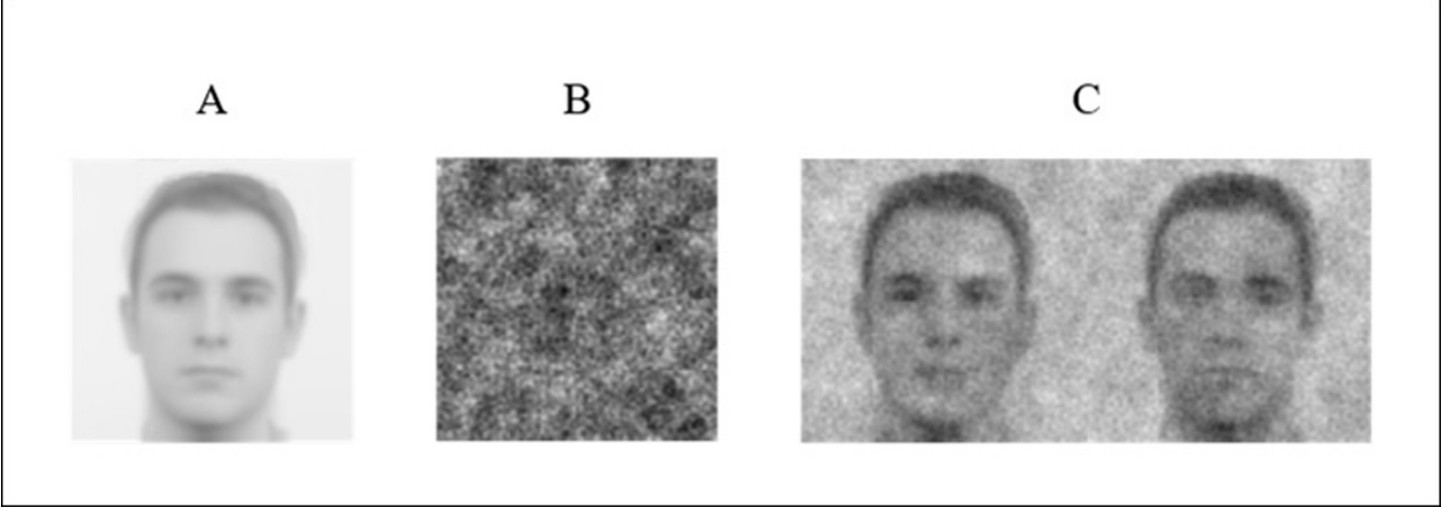

**Fig 1. Base image and noise pattern.** Base image (A), example of a random noise pattern (B), and an example of a pair of noisy faces produced by adding (left) or subtracting (right) the noise pattern from the base image (C).

friendliness (warm, sociable, and friendly) and morality (trustworthy, moral, honest) for the warmth dimension [11, 12]. The competence and warmth dimensions had adequate levels of reliability (all α's ≥ .77). In phase 3, we assessed judges' explicit impressions about the CIs by means of two traits, dominance and trustworthiness, whereas in phase 4 the images were rated on competence, warmth, dominance, and trustworthiness, using the same 7-point scale.

## Procedures

In phase 1, producers underwent an impression formation task, a RC task, and a rating task. The impression formation task, implemented in Psytoolkit [80, 81], was built upon Judd et al.'s [18] procedure in which participants form an impression by reading a series of behaviors allegedly performed by members of two fictitious groups (i.e., the "Green group" and the "Blue group"). We adapted the twenty-four behaviors diagnostic of the two fundamental dimensions from previous studies [18, 23, 82]. The high (low) group was ascribed six positive (negative) and two negative (positive) competence related behaviors, as well as two negative and two positive warmth related behaviors. Therefore, the high group was portrayed as more competent (i.e., the manipulated dimension) than the low group, but both were equally warm (i.e., the unmanipulated dimension). Group identity (i.e., "Blue" or "Green") was randomized across participants. Behaviors were paced one at a time for 6000ms with an inter-stimulus delay of 400ms. Each behavior was anonymized and presented with a colored background (i.e., blue or green) to signal group membership. As in Dotsch et al. [64] and to facilitate the forthcoming RC task, we randomly coupled each behavior with a noisy face of what we presented as a group member. On average, exemplar faces did not differ between target groups across participants. To solidify their impressions, participants went through the list of behaviors, once in random order, and once sorted by group identity. They then reported in a few lines their global impression of the two groups. Next, participants performed a 350 trials-RC-task with a single target group (i.e., "Blue" or "Green" group) being randomized between participants. Trials appeared in random order, and the position of the noisy faces within a pair (i.e., left vs. right) was randomized at each trial. Participants then rated the two groups on a series of traits encompassing the warmth and competence dimension. Traits appeared in

random order, with the identity of the first group being randomized across participants. Finally, participants provided some demographic information and were debriefed. Thereafter, a new sample of independent judges rated in Qualtrics the two average CIs (i.e., one for the high group, and one for the low group) presented next to each other on a series of traits related to warmth and competence. CIs' position and traits order were randomized. Finally, judges provided some demographic information and were debriefed.

In phase 2, we adopted the same general procedure, with three exceptions. First, we manipulated the warmth dimension (instead of competence) in the impression formation. Specifically, we presented behaviors in such a way that the high-warmth group was warmer but equally competent than the low group. Second, we did not provide exemplar faces of group members during the impression formation phase given that previous studies have shown that participants could form a visual representation of minimal groups even without seeing exemplar faces [65]. Third, we expanded our array of traits during the CIs' rating phase to tap the four facets of the two fundamental dimensions.

In phase 3, we randomly assigned judges to one of two conditions: They either rated the CIs produced in phase 1 (i.e., competence manipulated) or those in phase 2 (i.e., warmth manipulated). In each condition, both high and low CIs were adjacent, and judges rated them on dominance and trustworthiness. CIs' position and traits order were randomized. After the ratings, judges provided some demographic information and were debriefed.

Finally, in phase 4, we randomly assigned judges to one of six conditions: They either saw the CIs produced in phase 1 (i.e., competence manipulated) or those produced in phase 2 (i.e., warmth manipulated) and rated them either on the social dimensions (i.e., competence and warmth), the facial dimensions (i.e., dominance and trustworthiness), or both. In each condition, both high and low CIs were adjacent, and judges rated them on two or four traits: competence, warmth, dominance, and trustworthiness. The order of the CIs position and traits was randomized, and after giving their ratings, the judges provided some demographic information and received a debriefing.

## Designs

Phase 1 relied on a 2 (target group: high- vs. low-competence group) × 2 (judged dimension: competence vs. warmth) repeated measures design. Similarly, phase 2 consisted of a 2 (target group: high- vs. low-warmth group) × 2 (judged dimension: competence vs. warmth) repeated measures design. In phase 3, the design was a 2 (manipulated dimension: competence vs. warmth) × 2 (target group: high- vs. low group on the manipulated dimension) × 2 (judged dimension: dominance vs. trustworthiness), with the first factor varying between participants and the last two within them. Lastly, phase 4 was characterized by a 2 (design type: between-participants vs. within-participant) × 2 (manipulated dimension: competence vs. warmth) × 2 (target group: high- vs. low group on the manipulated dimension) × 2 (judged dimension: vertical vs. horizontal) × 2 (dimensional model: social vs. facial) design, with the first and the second factor varying between participants, the third and the fourth varying within them and the dimensional model factor varied both within and between them [83].

## Ethical statement

The authors declare that the research was conducted ethically, the results are reported honestly, the submitted work is original and not (self-)plagiarized, and authorship reflects individuals' contributions. This research did not include any minors, all participants agreed to an informed consent form (online) prior to their participation in the studies. Participants were debriefed post-participation, ensuring their well-being, and understanding of the study's

purpose. The PhD project was approved by the ethical committee from the Research Institute for Psychological Sciences at Université catholique de Louvain.

## Results

### Phase 1

We submitted phase 1 producers' ratings of the two target groups to a 2 (target group: high- vs. low-competence group) × 2 (judged dimension: competence vs. warmth) repeated measures ANOVA. There was a main effect of target group, such that the high group received higher ratings than the low group ($M_{high}$ = 3.91; $SD_{high}$ = 1.29; $M_{low}$ = 3.46; $SD_{low}$ = 1.37), $F(1,132)$ = 13.50, $p < .001$, $\eta_p^2$ = .09. There was also a main effect of judged dimension, such that the competence dimension was rated higher than the warmth dimension ($M_{competence}$ = 3.78; $SD_{competence}$ = 1.38; $M_{warmth}$ = 3.59; $SD_{warmth}$ = 1.31), $F(1,132)$ = 13.88, $p < .001$, $\eta_p^2$ = .09. Importantly, the target group × judged dimension interaction was significant, $F(1,132)$ = 52.07, $p < .001$, $\eta_p^2$ = .28. Confirming the success of our manipulation, the high group came across as more competent ($M_{high}$ = 4.40; $SD_{high}$ = 1.12; $M_{low}$ = 3.15; $SD_{low}$ = 1.34), $F(1,132)$ = 62.73, $p < .001$, $\eta_p^2$ = .32. Crucially, supporting compensation, the high group also appeared less warm than the low group ($M_{high}$ = 3.42; $SD_{high}$ = 1.26; $M_{low}$ = 3.77; $SD_{low}$ = 1.33), $F(1,132)$ = 4.18, $p = .043$, $\eta_p^2$ = .03.

We then assessed the producers' visual representation of the two social targets. To compute the classification image (CI) of the high and low-competence group, we averaged every noise pattern selected by the producers and superimposed it on the base-face (Fig 2).

We then submitted phase 1 judges' ratings of the two CIs captured from the producers to a 2 (target group: high- vs. low-competence group) × 2 (judged dimension: competence vs. warmth) two-way repeated measure ANOVA. There was a main effect of target group, such that the high-competence group CI was rated lower than the low-competence group CI ($M_{high}$ = 4.01; $SD_{high}$ = 1.49; $M_{low}$ = 4.83; $SD_{low}$ = 1.25), $F(1,54)$ = 36.57, $p < .001$, $\eta_p^2$ = .39. There was also a main effect of judged dimension, such that the competence dimension was rated higher than the warmth dimension ($M_{competence}$ = 4.60; $SD_{competence}$ = 1.28; $M_{warmth}$ = 4.24; $SD_{warmth}$ = 1.55), $F(1,54)$ = 13.17, $p < .001$, $\eta_p^2$ = .19. Importantly, the target group × judged dimension interaction proved significant, $F(1,54)$ = 72.15, $p < .001$, $\eta_p^2$ = .55. Specifically, judges rated the high-competence CI as more competent ($M_{high}$ = 4.77; $SD_{high}$ = 1.33; $M_{low}$ = 4.43; $SD_{low}$ = 1.22), although this difference was marginal, $F(1,54)$ = 3.41, $p = .070$, $\eta_p^2$ = .06. Crucially, and in line with the compensation effect, judges rated the high-competence CI less warm than the low-competence CI ($M_{high}$ = 3.24; $SD_{high}$ = 1.22; $M_{low}$ = 5.24; $SD_{low}$ = 1.15), $F(1,54)$ = 95.51, $p < .001$, $\eta_p^2$ = .62.

### Phase 2

We submitted phase 2 producers' ratings of the two target groups to a 2 (target group: high- vs. low-competence group) × 2 (judged dimension: competence vs. warmth) repeated measures ANOVA. There was a main effect of target group, such that the high group received higher ratings than the low group ($M_{high}$ = 4.07; $SD_{high}$ = 1.24; $M_{low}$ = 3.28; $SD_{low}$ = 1.43), $F(1,62)$ = 32.23, $p < .001$, $\eta_p^2$ = .33. There was also a main effect of judged dimension, such that the competence dimension was rated higher than the warmth dimension ($M_{competence}$ = 3.90; $SD_{competence}$ = 1.11; $M_{warmth}$ = 3.45; $SD_{warmth}$ = 1.60), $F(1,62)$ = 32.36, $p < .001$, $\eta_p^2$ = .33. The target group × judged dimension interaction was significant, $F(1,62)$ = 65.12, $p < .001$, $\eta_p^2$ = .50. In line with our manipulation, the high-warmth group came across as warmer ($M_{high}$ = 4.57; $SD_{high}$ = 1.15; $M_{low}$ = 2.32; $SD_{low}$ = 1.13), $F(1,62)$ = 89.33, $p < .001$, $\eta_p^2$ = .58. Importantly, the high group appeared less competent than the low-warmth group ($M_{high}$ = 3.56; $SD_{high}$ =

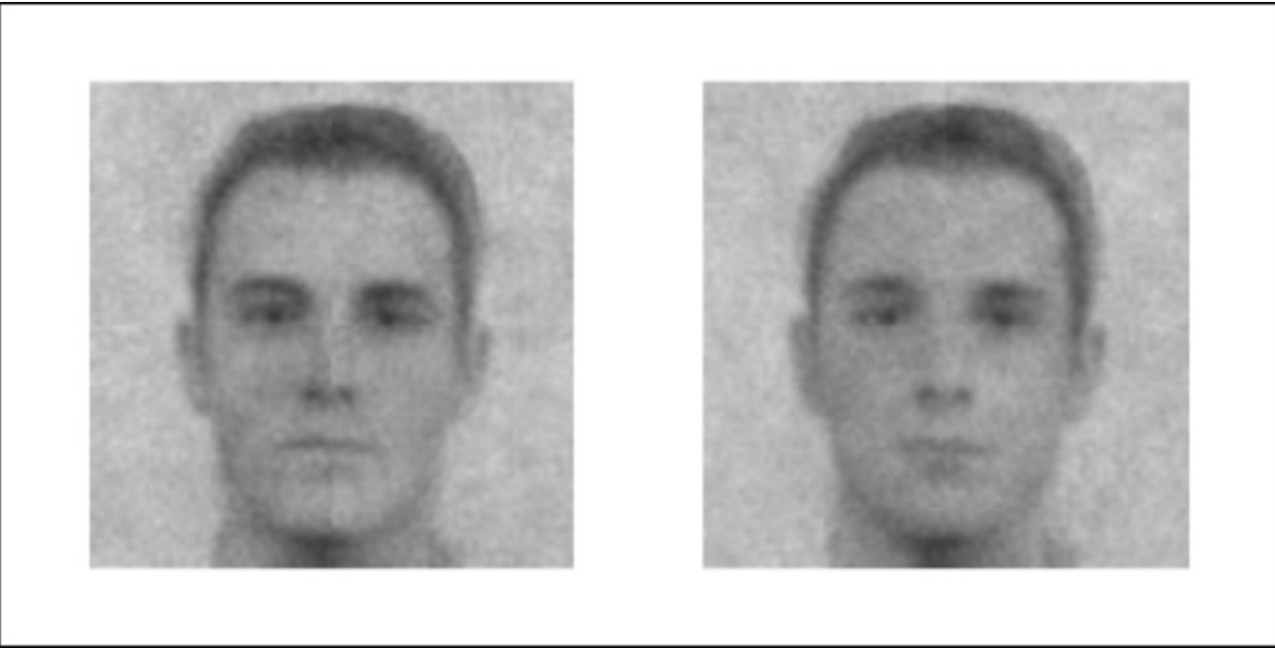

**Fig 2. High- and low-competence CIs.** High-competence group (left) and low-competence group (right) classification images (CIs). CIs were generated using the scaling constant method (c = .004).

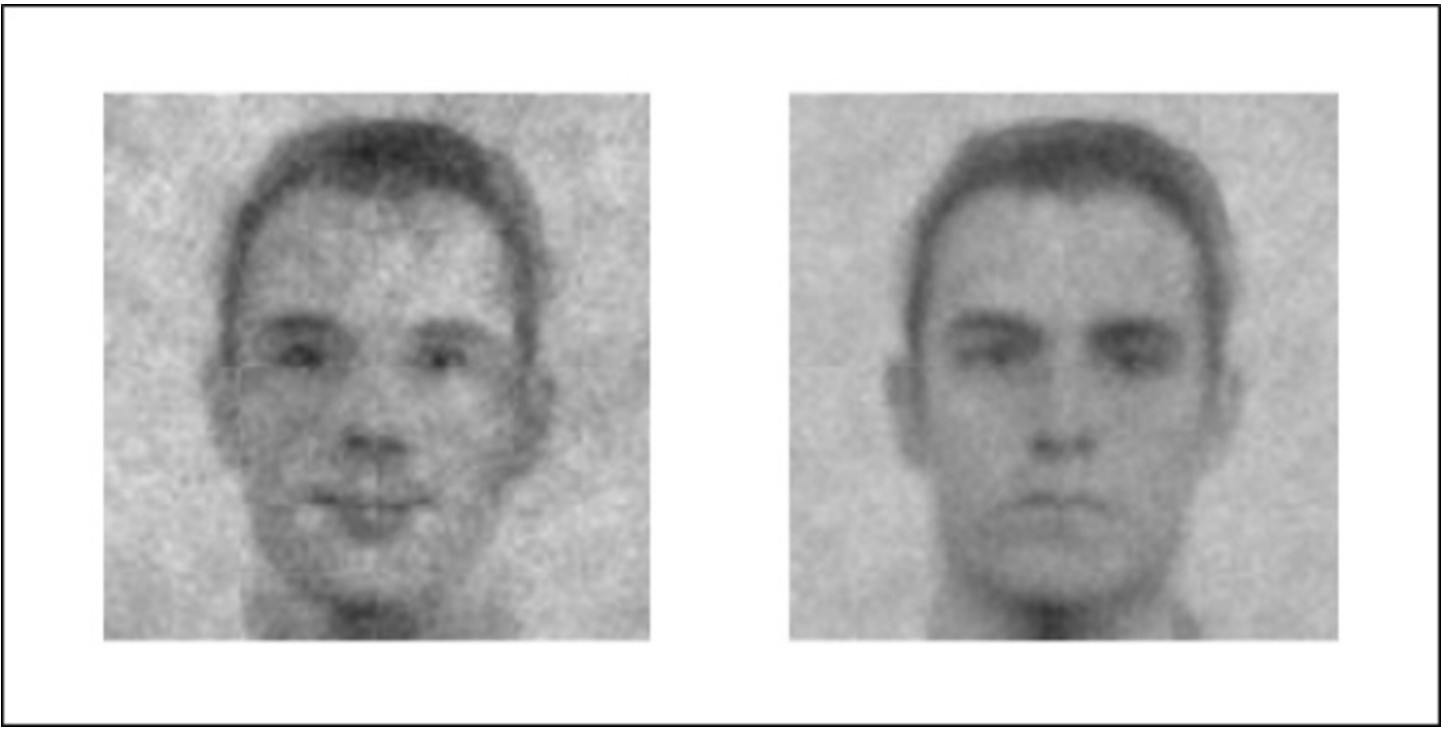

**Fig 3. High- and low-warmth CIs.** High-warmth group (left) and low-warmth group (right) classification images (CIs). CIs were generated using the scaling constant method (c = .006).

1.13; $M_{\text{low}}$ = 4.23; $SD_{\text{low}}$ = 0.99), $F(1,62)$ = 9.35, $p < .01$, $\eta^2_p$ = .13, thus supporting compensation.

Next, we assessed the producers' visual representation of the two social targets. To compute the classification image (CI) of the high and low-warmth group, we averaged every noise pattern selected by all the producers and superimposed it on the base image (Fig 3).

We submitted phase 2 judges' ratings of the two CIs generated by the producers to a 2 (target group: high- vs. low-competence group) × 2 (judged dimension: competence vs. warmth) repeated measures ANOVA. There was a main effect of target group, such that the high group CI was rated higher than the low group CI ($M_{\text{high}}$ = 4.65; $SD_{\text{high}}$ = 1.08; $M_{\text{low}}$ = 3.58; $SD_{\text{low}}$ = 1.13), $F(1,101)$ = 80.90, $p < .001$, $\eta_p^2$ = .44. There was also a main effect of judged dimension, such that the competence dimension was rated higher than the warmth dimension ($M_{\text{competence}}$ = 4.21; $SD_{\text{competence}}$ = 0.96; $M_{\text{warmth}}$ = 4.02; $SD_{\text{warmth}}$ = 1.44), $F(1,101)$ = 23.62, $p < .001$, $\eta_p^2$ = .19. Importantly, the target group × judged dimension interaction proved significant, $F(1,101)$ = 149.77, $p < .001$, $\eta_p^2$ = .59. Follow-up tests revealed that the high-warmth CI came across as warmer ($M_{\text{high}}$ = 5.02; $SD_{\text{high}}$ = 1.07; $M_{\text{low}}$ = 3.01; $SD_{\text{low}}$ = 1.00), $F(1,101)$ = 177.60, $p < .001$, $\eta^2_p$ = .63, but not significantly more competent than the low-warmth CI ($M_{\text{high}}$ = 4.27; $SD_{\text{high}}$ = 0.96; $M_{\text{low}}$ = 4.15; $SD_{\text{low}}$ = 0.96), $F(1,101)$ = 0.80, $p = .373$, $\eta^2_p$ = .01. These results replicated when conducting the analyses by taking into account the facets.

## Phase 3

We submitted phase 3 judges' ratings of the CIs to a 2 (manipulated dimension: competence vs. warmth) × 2 (target group: high- vs. low group on the manipulated dimension) × 2 (judged dimension: dominance vs. trustworthiness) mixed model analysis, with the first factor varying between participants and the last two within them. There was no main effect of manipulated dimension, such that the ratings did not significantly differ when the manipulated dimension was competence or warmth ($M_{\text{competence}}$ = 4.15; $SD_{\text{competence}}$ = 1.42; $M_{\text{warmth}}$ = 4.01; $SD_{\text{warmth}}$ = 1.62), $F(1,194)$ = 2.18, $p = .141$, $\eta_p^2$ = .01. There was no main effect of target group, such that the high group ratings were not significantly different from the low group ($M_{\text{high}}$ = 4.05; $SD_{\text{high}}$ = 1.47; $M_{\text{low}}$ = 4.11; $SD_{\text{low}}$ = 1.57), $F(1,194)$ = 0.84 $p = .360$, $\eta_p^2 < .01$. There was a main effect of judged dimension, such that the dominance dimension was rated higher than the trustworthiness dimension ($M_{\text{trustworthyness}}$ = 3.89; $SD_{\text{trustworthyness}}$ = 1.48; $M_{\text{dominance}}$ = 4.27; $SD_{\text{dominance}}$ = 1.55), $F(1,194)$ = 26.80, $p < .001$, $\eta_p^2$ = .12. The manipulated dimension × target group was significant, $F(1,194)$ = 11.92, $p < .001$, $\eta_p^2$ = .06, as was the target group × judged dimension, $F(1,194)$ = 7.71, $p < .01$, $\eta_p^2$ = .04. The manipulated dimension × judged dimension did not reach significance, $F(1,194)$ = 0.10, $p = .757$, $\eta_p^2 < .01$.

Importantly, the manipulated dimension × target group × judged dimension interaction proved significant, $F(1,194)$ = 314.56, $p < .001$, $\eta_p^2$ = .61. Specifically, the target group × judged dimension was significant when competence was the manipulated dimension, $F(1,194)$ = 112.45, $p < .001$, $\eta_p^2$ = .36. Follow-up analyses confirmed that the high-competence CI came across as more dominant ($M_{\text{high}}$ = 5.19; $SD_{\text{high}}$ = 1.17; $M_{\text{low}}$ = 3.52; $SD_{\text{low}}$ = 1.25), $F(1,194)$ = 97.05, $p < .001$, $\eta^2_p$ = .33, but also, in line with compensation, less trustworthy than the low-competence CI, ($M_{\text{high}}$ = 3.29; $SD_{\text{high}}$ = 1.12; $M_{\text{low}}$ = 4.61; $SD_{\text{low}}$ = 1.22), $F(1,194)$ = 61.16, $p < .001$, $\eta^2_p$ = .24. Also, the target group × judged dimension was significant when warmth was the manipulated dimension, $F(1,194)$ = 209.34, $p < .001$, $\eta_p^2$ = .51, such that judges rated the high-warmth CI more trustworthy than the low-warmth CI ($M_{\text{high}}$ = 4.71; $SD_{\text{high}}$ = 1.34; $M_{\text{low}}$ = 2.95; $SD_{\text{low}}$ = 1.35), $F(1,194)$ = 106.56, $p < .001$, $\eta^2_p$ = .35, and, supporting compensation, as less dominant ($M_{\text{high}}$ = 3.02; $SD_{\text{high}}$ = 0.97; $M_{\text{low}}$ = 5.36; $SD_{\text{low}}$ = 1.24), $F(1,99)$ = 190.17, $p < .001$, $\eta^2_p$ = .49.

## Phase 4

We submitted phase 4 judges' ratings of the CIs to a 2 (design type: between-participants vs. within-participant) × 2 (manipulated dimension: competence vs. warmth) × 2 (target group: high- vs. low group on the manipulated dimension) × 2 (judged dimension: vertical vs. horizontal) × 2 (dimensional model: social vs. facial) mixed model analysis Because the regression analysis involved 32 terms, for the sake of clarity and conciseness, we only report here the statistical tests of the regression terms required to test our hypothesis. The complete statistical analyses are publicly available on the OSF repository.

We conducted the analysis with linear mixed models [84]. As we hoped, the five-way interaction was far from significant, $F(1,1296.99) = 0.01$, $p = .926$, suggesting that the specific design did not moderate the obtained pattern of judgments. In sharp contrast, and consistent with findings from phases 1–3, the manipulated dimension × target group × judged dimension × dimensional model proved significant, $F(1,1296.99) = 27.11$, $p < .001$.

Consistent with the results found in phases 1 and 2, follow-up analyses revealed that the manipulated dimension × target group × judged dimension was significant for the social model, $F(1,1296.99) = 132.53$, $p < .001$. As expected, the target group × judged dimension was significant when competence was the manipulated dimension (i.e., phase 1), $F(1,1296.99) = 37.90$, $p < .001$. Specifically, there was no significant difference on competence between the high- and low-competence CIs ($M_{high} = 4.34$; $SD_{high} = 1.12$; $M_{low} = 4.26$; $SD_{low} = 1.03$), $F(1,1296.99) = 0.23$, $p = .633$, whereas the latter was perceived as warmer ($M_{high} = 3.01$; $SD_{high} = 1.27$; $M_{low} = 4.39$; $SD_{low} = 1.48$), $F(1,1296.99) = 67.1$, $p < .001$. Conversely, the target group × judged dimension was significant when warmth was the manipulated dimension (i.e., phase 2), $F(1,1296.99) = 102.30$, $p < .001$. This time, the high-warmth CI was perceived as warmer than the low-warmth CI ($M_{high} = 4.90$; $SD_{high} = 1.45$; $M_{low} = 2.38$; $SD_{low} = 1.16$), $F(1,1296.99) = 222.72$, $p < .001$, but there was no significant difference on competence between the two CIs ($M_{high} = 4.11$; $SD_{high} = 1.20$; $M_{low} = 4.01$; $SD_{low} = 1.24$), $F(1,1296.99) = 0.38$, $p = .535$.

Turning to the facial dimensional model, the manipulated dimension × target group × judged dimension proved significant, $F(1,1296.99) = 362.01$, $p < .001$. As expected, the target group × judged dimension was significant when competence was the manipulated dimension (i.e., phase 1), $F(1,1296.99) = 127.38$, $p < .001$, such that the high-competence CI was perceived as more dominant ($M_{high} = 5.27$; $SD_{high} = 1.10$; $M_{low} = 3.49$; $SD_{low} = 1.21$), $F(1,1296.99) = 114.10$, $p < .001$, but less trustworthy than the low-competence CI ($M_{high} = 3.43$; $SD_{high} = 1.16$; $M_{low} = 4.31$; $SD_{low} = 1.38$), $F(1,1296.99) = 27.87$, $p < .001$. Conversely, the target group × judged dimension was significant when warmth was the manipulated dimension (i.e., phase 2), $F(1,1296.99) = 244.36$, $p < .001$, such that the high-warmth CI was perceived as less dominant ($M_{high} = 3.11$; $SD_{high} = 1.28$; $M_{low} = 5.36$; $SD_{low} = 1.33$), $F(1,1296.99) = 185.23$, , $p < .001$, but more trustworthy than the low-warmth CI ($M_{high} = 4.32$; $SD_{high} = 1.46$; $M_{low} = 2.91$; $SD_{low} = 1.01$), $F(1,1296.99) = 72.19$, $p < .001$.

## Discussion

### Aims of the present research

Although robust, research on the compensation effect has exclusively been built on verbal outputs such as written descriptions or reactions to group labels. Furthermore, the designs and measures used in previous studies restrained participants' responses to a limited set of options determined *a priori* by researchers (e.g., [18, 24, 25, 85]). These two aspects are not trivial limitations when considering that everyday life social impressions are formed freely and comprise multi-modal components (e.g., auditory stereotypes [86]; visual stereotypes [64]). The current

effort addressed these limitations by testing the compensation effect in a visual and uncon-
strained manner. All ethical guidelines and standards were rigorously adhered to in the present
research, ensuring the integrity and ethical compliance of the study.

### Visual correlates of the compensation effect

In line with findings showing that top-down beliefs and stereotypes can affect the way we pic-
ture others in our mind [3, 26, 68], our results indicate that the compensation effect does also
bias the visual template of social targets. Interestingly, however, this occurs on the horizontal
dimensions of either trustworthiness or warmth and on the vertical dimension of dominance
but not on the vertical dimension of competence [33].

In phase 1, we tested the visual correlates of the compensation effect. To do so, we captured
both participants' (producers) verbal (i.e., traits ratings) and visual (i.e., CI–classification
image) impressions of two fictitious groups that initially differed on competence but were
equal on warmth. In line with previous findings, our results revealed the presence of a clear-
cut compensation pattern among producers' verbal ratings. However, their facial prototypes of
the two groups contrasted strongly in terms of warmth but only marginally in terms of compe-
tence. This outcome raised the possibility that the sample of traits tapping the competence
dimension used by the judges to rate the visual images did not adequately capture this dimen-
sion. Indeed, it only encompassed one of its two facets, namely ability. Moreover, the question
remained whether this visual compensation pattern would hold when the initial difference
between the groups resides on warmth rather than competence.

In phase 2, we extended these findings by manipulating the dimension of warmth instead
of competence (i.e., one group was initially presented as warmer but equally competent than
the second group). It is noteworthy that this manipulation constituted a more conservative test
because compensation is reportedly weaker under such circumstances [9, 18, 24]. Moreover,
we enlarged the spectrum of traits used to rate the visual presentations to the four facets of the
two fundamental dimensions. Again, producers compensated on their rating of the two tar-
gets, but the evaluation of their classification images (CIs) by independent judges failed to
show a compensation. Specifically, the high-warmth target was judged as warmer than the
low-warmth target, but they did not significantly differ in terms of competence. These findings
were intriguing since, in both phases, producers' ratings clearly acknowledged a competence
gap between the two social targets. Building on existing literature [50] suggesting that cues of
dominance (e.g., physical strength, aggressivity, or masculinity) may be more readily encoded
into facial features than cues of competence (e.g., intelligence, status, or skills), we conjectured
that the perceived gap in competence expressed in the verbal ratings could have been better
translated in the face space as dominance.

In line with this hypothesis, phase 3 revealed a strong compensation pattern when re-evalu-
ating the CIs from phase 1 and 2, but this time on the facial dimensions of dominance and
trustworthiness. Again, competence failed to reveal a compensatory pattern in judges' ratings
of the target faces. This finding adds to the growing body of literature indicating that the two
vertical dimensions seem to work in rather distinct ways (e.g., [13, 50, 71–73]). A plausible
explanation may be that the producers visually encoded competence as dominance rather than
competence given that the latter is more concrete in terms of facial cues [50].

Finally, we sought to replicate and further ascertain these results in phase 4 by manipulating
the experimental design (i.e., within-participants vs. between-participants), and tested whether
the social dimension of competence could be sharply differentiated from the facial dimension
of dominance even when measured at the same time. Specifically, judges either rated the CIs
on the social dimensions (i.e., competence and warmth), the facial dimensions (i.e.,

dominance and trustworthiness), or both. In line with our previous findings, results showed that the pattern upheld whether the vertical dimension of dominance was gauged along with or independently of the vertical dimension of competence.

### Future research directions and limitations

Although our work focused on the evaluations made by human judges, future investigations may want to test for the presence of a compensatory pattern in the produced CIs by examining the amount of physical information associated with each dimension (warmth, competence, trustworthiness, and dominance) present in these visual renderings. That is, by examining correlation of the pixels' luminance (see [68]; e.g., [58, 66, 70, 72]) found in the CIs generated in a compensatory setting and in those independently generated CIs diagnostic of each dimension (e.g., competence).

A limitation of the present research is that the CIs were computed and analyzed at the group level (i.e., averaged visual representation from all the participants within a given experimental condition). Indeed, average CIs do not capture the inter-individual variability and do not allow for more fine-grained analyses ([65, 68]; see [87]). For instance, it may be interesting for future research to test the degree to which producers' verbal compensatory ratings correlate (or not) with the evaluation by external judges of their idiosyncratic classification images (CIs). Another caveat is that we did not assess producers' verbal ratings of the two target groups on the facial dimensions, as it was not initially the scope of the current research. Although our manipulation relied on behaviors diagnostic of competence and warmth (see [18, 23, 82]), they may also be informative in terms of dominance and trustworthiness, and thus account for the presence of dominance and trustworthiness in the CIs.

## Conclusions

The present research is the first to provide direct experimental evidence for the visual correlates of the compensation effect, thus highlighting its pervasiveness beyond verbal measures. Clearly, our work opens new avenues to investigate compensation in a way that is more germane to really life social interactions, that is, under near spontaneous and unconstrained conditions.

## Author Contributions

**Conceptualization:** Mathias Schmitz, Antoine Vanbeneden, Vincent Yzerbyt.

**Data curation:** Mathias Schmitz, Antoine Vanbeneden.

**Formal analysis:** Mathias Schmitz, Antoine Vanbeneden.

**Investigation:** Mathias Schmitz, Antoine Vanbeneden, Vincent Yzerbyt.

**Methodology:** Mathias Schmitz, Antoine Vanbeneden.

**Project administration:** Vincent Yzerbyt.

**Supervision:** Vincent Yzerbyt.

**Validation:** Vincent Yzerbyt.

**Writing – original draft:** Mathias Schmitz, Antoine Vanbeneden, Vincent Yzerbyt.

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
