## [Decision Letter · Decision Letter 0]

8 Jan 2023

PONE-D-22-10567The many faces of compensation: The similarities and differences between social and facial models of perceptionPLOS ONE

Dear Dr. Schmitz,

Thank you for submitting your manuscript to PLOS ONE. After careful consideration, we feel that it has merit but does not fully meet PLOS ONE’s publication criteria as it currently stands. Therefore, we invite you to submit a revised version of the manuscript that addresses the points raised during the review process.

We look forward to receiving your revised manuscript.

Kind regards,

Xiaowei Li

Academic Editor

PLOS ONE

Journal Requirements:

3. Please change "female” or "male" to "woman” or "man" as appropriate, when used as a noun (see for instance https://apastyle.apa.org/style-grammar-guidelines/bias-free-language/gender).

6. We note that Figures 1-3 includes an image of a [patient / participant / in the study].

Reviewers' comments:

Reviewer's Responses to Questions

**Comments to the Author**

1. Is the manuscript technically sound, and do the data support the conclusions?

Reviewer #1: Partly

Reviewer #2: Yes

2. Has the statistical analysis been performed appropriately and rigorously? 

Reviewer #1: I Don't Know

Reviewer #2: Yes

3. Have the authors made all data underlying the findings in their manuscript fully available?

Reviewer #1: Yes

Reviewer #2: Yes

4. Is the manuscript presented in an intelligible fashion and written in standard English?

Reviewer #1: Yes

Reviewer #2: Yes

5. Review Comments to the Author

Reviewer #1: Well done on a very interesting topic. Several amendments need to be done on the overall formatting. I reckon all experiments should be merge together and not partly discussed- methods, results, discussion- in order to have an easy understanding on the study.

Reviewer #2: Overall, this is an interesting article.

Through 4 experiments, the authors demonstrate that the expectations of the facial content of two novel groups that differed on one of the two social dimensions are biased in a compensatory manner on the facial dimensions of trustworthiness, warmth, and dominance.

But the layout of the author's manuscript is more like a thesis than a research article. Manuscripts should be organized as requested by Plos One.

In addition, the manuscript should have some streamlining of the content of the article to make it more readable while undergoing formatting revisions.

6. PLOS authors have the option to publish the peer review history of their article (what does this mean?). If published, this will include your full peer review and any attached files.

Reviewer #1: No

Reviewer #2: No

---

## [Author Response · Author response to Decision Letter 0]

30 Aug 2023

Dear Editor,

We are pleased to submit the revised version of our original research manuscript, "The many faces of compensation: The similarities and differences between social and facial models of perception". We appreciate the opportunity to address the formatting issues and incorporate the valuable suggestions provided by the reviewers.

We have made significant changes to the manuscript to improve its cohesiveness and accessibility to readers. We have merged all experiments together into a single research program with four phases, as well as streamlined the methods, results, and discussion to enhance the overall understanding of our study. Moreover, we have formatted the manuscript in accordance to PLOS ONE guidelines.

We confirm that we have no conflict of interest, and the results reported in this manuscript are original and previously unpublished. All authors have approved the revised version of the manuscript.

We thank you for considering our revised submission and look forward to hearing your editorial decision.

Sincerely,

Mathias Schmitz, Antoine Vanbeneden, &Vincent Yzerbyt

 

Journal Requirements:

Response: We ensured that we meet all of PLOS ONE's style requirements, including file naming conventions, for our revised submission.

Response: Information about informed consent is provided in both the Method and the Ethical statement sections.

3. Please change "female” or "male" to "woman” or "man" as appropriate, when used as a noun (see for instance https://apastyle.apa.org/style-grammar-guidelines/bias-free-language/gender).

Response: The manuscript was updated accordingly.

Response: We made sure that the grant numbers matched.

Response: We included the full ethics statement in the Methods and also incorporated the name of the ethics committee that approved the study. We also mention that participants filled in an online informed consent prior to taking part in the research.

6. We note that Figures 1-3 includes an image of a [patient / participant / in the study].

Response: We mentioned that the base image used in the reverse correlations tasks was constructed from six men's faces selected from the Radboud Face Database for which we obtain permission of use. Furthermore, all images presented in Figure 1-3 were artificially constructed.

Reviewers' comments:

Reviewer's Responses to Questions

Comments to the Author

1. Is the manuscript technically sound, and do the data support the conclusions?

Reviewer #1: Partly

Reviewer #2: Yes

2. Has the statistical analysis been performed appropriately and rigorously?

Reviewer #1: I Don't Know

Reviewer #2: Yes

3. Have the authors made all data underlying the findings in their manuscript fully available?

Reviewer #1: Yes

Reviewer #2: Yes

4. Is the manuscript presented in an intelligible fashion and written in standard English?

Reviewer #1: Yes

Reviewer #2: Yes

5. Review Comments to the Author

Reviewer #1: Well done on a very interesting topic. Several amendments need to be done on the overall formatting. I reckon all experiments should be merge together and not partly discussed- methods, results, discussion- in order to have an easy understanding on the study.

Reviewer #2: Overall, this is an interesting article.

Through 4 experiments, the authors demonstrate that the expectations of the facial content of two novel groups that differed on one of the two social dimensions are biased in a compensatory manner on the facial dimensions of trustworthiness, warmth, and dominance.

But the layout of the author's manuscript is more like a thesis than a research article. Manuscripts should be organized as requested by Plos One.

In addition, the manuscript should have some streamlining of the content of the article to make it more readable while undergoing formatting revisions.

6. PLOS authors have the option to publish the peer review history of their article (what does this mean?). If published, this will include your full peer review and any attached files.

Do you want your identity to be public for this peer review? For information about this choice, including consent withdrawal, please see our Privacy Policy.

Reviewer #1: No

Reviewer #2: No

Response to reviewers: Thank you to the reviewers for taking the time to provide feedback on our article. We appreciate your insights and suggestions and have addressed your requests accordingly.

We agree that the formatting of our article needed improvement. As such, we have merged all experiments together into a single research divided into four phases. We also streamlined the discussion of methods, results, and discussion to improve the overall understanding of our study. Additionally, we have organized our manuscript as requested by PLOS ONE and streamlined the content to make it more readable.

---

## [Decision Letter · Decision Letter 1]

24 Sep 2023

PONE-D-22-10567R1The many faces of compensation: The similarities and differences between social and facial models of perceptionPLOS ONE

Dear Dr. Schmitz,

Thank you for submitting your manuscript to PLOS ONE. After careful consideration, we feel that it has merit but does not fully meet PLOS ONE’s publication criteria as it currently stands. Therefore, we invite you to submit a revised version of the manuscript that addresses the points raised during the review process.

We look forward to receiving your revised manuscript.

Kind regards,

Xiaowei Li

Academic Editor

PLOS ONE

Reviewers' comments:

Reviewer's Responses to Questions

**Comments to the Author**

1. If the authors have adequately addressed your comments raised in a previous round of review and you feel that this manuscript is now acceptable for publication, you may indicate that here to bypass the “Comments to the Author” section, enter your conflict of interest statement in the “Confidential to Editor” section, and submit your "Accept" recommendation.

Reviewer #1: All comments have been addressed

Reviewer #2: (No Response)

2. Is the manuscript technically sound, and do the data support the conclusions?

Reviewer #1: Yes

Reviewer #2: Yes

3. Has the statistical analysis been performed appropriately and rigorously? 

Reviewer #1: I Don't Know

Reviewer #2: Yes

4. Have the authors made all data underlying the findings in their manuscript fully available?

Reviewer #1: Yes

Reviewer #2: Yes

5. Is the manuscript presented in an intelligible fashion and written in standard English?

Reviewer #1: Yes

Reviewer #2: Yes

6. Review Comments to the Author

Reviewer #1: (No Response)

Reviewer #2: This manuscript tackles an important topic in the domain of social psychology, focusing on the compensation effect in visual impressions, which appears to be a novel angle.

Areas for Improvement:

Clarify Terminology: The use of jargon and specialized terms (e.g., "CI") without adequate initial definition can be a barrier for readers unfamiliar with the specific nuances of the topic. Ensure that all terms are defined upon first mention.

Streamline Content: Some sections have redundancy. Reducing repeated content can make the manuscript more concise and engaging.

Enhance Accessibility: The manuscript, especially the discussion, is dense in places. Consider simplifying complex sentences and breaking down intricate ideas for broader readership.

Citation Format: There are placeholders for citations throughout. Ensure they are replaced with proper references in a consistent format.

Enhanced Structure: Consider using subheadings more extensively to provide a clearer roadmap for readers, especially in the methods and discussion sections.

Ethical Considerations: While it's not clear from the presented sections, make sure ethical considerations related to the study's execution are explicitly mentioned elsewhere in the manuscript.

The manuscript presents a well-researched, detailed examination of the compensation effect in visual impressions. Given its novel approach and thorough exploration, it holds significant value to the field. With some refinements, particularly in enhancing clarity, streamlining content, and ensuring all terms and concepts are defined, the manuscript appears to be of publishable quality. The considerations highlighted above should strengthen the manuscript, making it even more compelling for publication.

7. PLOS authors have the option to publish the peer review history of their article (what does this mean?). If published, this will include your full peer review and any attached files.

Reviewer #1: No

Reviewer #2: No

---

## [Author Response · Author response to Decision Letter 1]

20 Dec 2023

See response_to_reviewers.docx

---

## [Decision Letter · Decision Letter 2]

15 Jan 2024

The many faces of compensation: The similarities and differences between social and facial models of perception

PONE-D-22-10567R2

Dear Dr. Schmitz,

We’re pleased to inform you that your manuscript has been judged scientifically suitable for publication and will be formally accepted for publication once it meets all outstanding technical requirements.

Kind regards,

Xiaowei Li

Academic Editor

PLOS ONE

Additional Editor Comments (optional):

Reviewers' comments:

Reviewer's Responses to Questions

**Comments to the Author**

1. If the authors have adequately addressed your comments raised in a previous round of review and you feel that this manuscript is now acceptable for publication, you may indicate that here to bypass the “Comments to the Author” section, enter your conflict of interest statement in the “Confidential to Editor” section, and submit your "Accept" recommendation.

Reviewer #2: All comments have been addressed

2. Is the manuscript technically sound, and do the data support the conclusions?

Reviewer #2: Yes

3. Has the statistical analysis been performed appropriately and rigorously? 

Reviewer #2: Yes

4. Have the authors made all data underlying the findings in their manuscript fully available?

Reviewer #2: Yes

5. Is the manuscript presented in an intelligible fashion and written in standard English?

Reviewer #2: Yes

6. Review Comments to the Author

Reviewer #2: Based on my review, the authors manuscript appears to be well-organized and focused on a relevant and interesting topic in the field of social perception. The structure follows conventional academic standards, with clear sections for Introduction, Methods, Results, and Discussion.

7. PLOS authors have the option to publish the peer review history of their article (what does this mean?). If published, this will include your full peer review and any attached files.

Reviewer #2: No

---

## [Editor Report · Acceptance letter]

16 Feb 2024

PONE-D-22-10567R2 

PLOS ONE

Dear Dr. Schmitz, 

I'm pleased to inform you that your manuscript has been deemed suitable for publication in PLOS ONE. Congratulations! Your manuscript is now being handed over to our production team.

Kind regards, 

on behalf of

Dr. Xiaowei Li 

Academic Editor

PLOS ONE